# Burden among informal caregivers of individuals with heart failure: A mixed methods study

**Angela Durante**[1,2]*, **Ahtisham Younas**[3], **Angela Cuoco**[4], **Josiane Boyne**[5], **Bridgette M. Rice**[6], **Raul Juarez-Vela**[1], **Valentina Zeffiro**[4], **Ercole Vellone**[4]

**1** Pre-department Unit of Nursing, GRUPAC, University of La Rioja, Logroño, Spain, **2** University of Eastern Piedmont, Novara, Italy, **3** Faculty of Nursing, Memorial University of Newfoundland, St. John's, Canada, **4** Department of Biomedicine and Prevention, University of Rome "Tor Vergata", Rome, Italy, **5** Department of Cardiology, Maastricht University Medical Center, Maastricht, The Netherlands, **6** M. Louise Fitzpatrick College of Nursing, Villanova University, Villanova, Philadelphia, United States of America

* angela.durante@uniupo.it

**Data Availability Statement:** Data cannot be shared publicly because of ethical restrictions. The transcripts are in a state in which data contain

## Abstract

### Aims

To develop a comprehensive understanding of caregiver burden and its predictors from a dyadic perspective.

### Method

A convergent mixed methods design was used. This study was conducted in three European countries, Italy, Spain, and the Netherlands. A sample of 229 HF patients and caregivers was enrolled between February 2017 and December 2018 from the internal medicine ward, outpatient clinic, and private cardiologist medical office. In total, 184 dyads completed validated scales to measure burden, and 50 caregivers participated in semi-structured interviews to better understand the caregiver experience. The Care Dependency Scale, Montreal Cognitive Assessment, and SF-8 Health Survey were used for data collection. Multiple regression analysis was conducted to identify the predictors and qualitative content analysis was performed on qualitative data. The results were merged using joint displays.

### Results

Caregiver burden was predicted by the patient's worse cognitive impairment, lower physical quality of life, and a higher care dependency perceived by the caregivers. The qualitative and mixed analysis demonstrated that caregiver burden has a physical, emotional, and social nature.

### Conclusions

Caregiver burden can affect the capability of informal caregivers to support and care for their relatives with heart failure. Developing and evaluating individual and community-

potentially identifying or sensitive patient information (like where they live or who their GP is). Please contact Professor Dr. Marco Di Nitto (marco.dinitto@uniroma2.eu) who has access to the data at the department of Biomedicine and Prevention of Rome Tor Vergata University where data are stored.

**Funding:** The first author was funded by the Heart Failure Association with the Nursing Training Fellowship Award. The project received funding also from the CECRI Centre of Excellence for Nursing Scholarship in Rome (2.12.5). The funders had no role in study design, data collection and analysis, decision to publish, or preparation of the manuscript.

**Competing interests:** The authors have declared that no competing interests exist.

based strategies to address caregiver burden and enhance their quality of life are warranted.

## Introduction

Heart failure (HF) is a chronic syndrome with a major impact on the world public health [1, 2]. Individuals with HF have to adhere to a strict and lifelong regimen consisting of self-care behaviours such as physical activity; monitoring for signs and symptoms; and taking medication, averaging 13 per day [3]; to achieve a balance and avoid exacerbations. Because of the complexity of self-care behaviours, people with advanced HF often rely on an informal caregiver who supports patients in managing drug therapy, daily weight monitoring, enforcing dietary prescriptions, and encouraging patient participation in physical activity [4]. In fact, in term of mutuality, the HF is often reframed as 'our problem' by caregivers rather than just the patient's problem [5]. Even if rewarding, in terms of personal and patients' well-being, caregivers' activities can be emotionally and physically intense, creating a situation where caregiver burden is common [6]. In HF patients, a higher caregiver burden can lead to reduced quality of care provided and reduced patient health.

A recent review about dyadic dynamics in HF showed the interrelationship of patient and caregiver and highlighted the need to include both care partners [7]. Despite caregivers play a pivotal role in the recovery of their relatives with HF, supporting them in promoting health and symptom management is challenging.

Even if evidence confirming the benefits to the patient of having a caregiver, the caregivers tend to experience great physical and emotional turmoil which affects their daily living [8]. The caregiver's load produces various repercussions, which can lead to the onset of sleep disorders, depression, anxiety, asthenia, gastrointestinal disorders, headaches, and even compromise the functioning of some aspects of daily life, such as personal space, role work, relationships, and family dynamics [9]. These issues had also arisen when the chronic illness progresses in caregivers of advanced HF patients, who claimed that they still lack adequate palliative care and communication concerning prognosis and end-of-life care [10].

Caregiver burden is often associated with negative outcomes like the dependency of the HF patient, cognitive impairment and poor functional status [11].

In previous recent studies [2, 6, 12–14], caregiver burden was associated with the care dependency of the HF patient, their cognitive impairment, health-related quality of life, and perceived social support and functional status.

In the existing quantitative research available there are no other cross-cultural studies. Furthermore, as the worldwide epidemic of heart failure grow [15], there is a dearth and a need of mixed-method studies exploring the nature of caregiver burden and any additional predictors that may affect the caregiver burden. Additionally, even if in the last decades awareness has been gained about the role of informal caregivers, there is still a dearth of cross-cultural studies exploring the nature of caregiver burden and any additional predictors that may affect it. Therefore, this mixed methods study was designed to address the qualitative and quantitative research gap [15] and generate a broader understanding of caregiver burden by honing onto the strengths both qualitative and quantitative methods.

The overarching aim was to develop a comprehensive understanding of caregiver burden and its predictors from a dyadic perspective. The specific objectives for each phase were: 1) To determine the levels of caregiver burden and its related variables from a dyadic perspective and to determine their relationship (Quantitative) and 2) To explore factors affecting caregiver

burden from the perspectives of caregivers and their relatives with HF (Qualitative). The mixed methods purpose was to examine the extent to which the quantitative factors affecting caregiver burden are consistent with or divergent from the qualitative findings from dyadic interviews.

## Materials and methods

This is a secondary analysis of a multi-site convergent mixed methods study, involving a parallel collection of quantitative and qualitative data to hone the strengths of these approaches and examine the consistencies across both datasets [16].

### Setting and sample

This study was conducted in three European countries, Italy, Spain, and the Netherlands. A sample of 229 caregivers was enrolled between February 2017 and December 2018 from the internal medicine ward, outpatient clinic, and private cardiologist medical office. A convenience sampling was used. The inclusion criteria were: being identified as informal caregiver —inside or outside the family—who provided most of the informal care for the patients, (b) being an informal caregiver for a patient who had been diagnosed with HF for at least three months before data collection (a clinical diagnosis of HF was based on the guidelines of the European Society of Cardiology) [1]. Semi-structured interviews were conducted exclusively to caregivers [17], without the presence of the patient, using an interview guide. Some of the questions used to explore the difficulties of the caregiving experience such as *"Do you have any problems in caring for -patient's name-?"* then opened up to insights into the mental health of the caregiver such as *"Have you experienced any emotional or psychological problems (such as stress, anxiety or loneliness, depression or moodiness)?"*. All interviews were audio-recorded then transcribed verbatim according to pure verbatim protocol. Further methodological details can be found in the parent study [18, 19].

### Data collection instruments

Burden is defined as the self-perception of psychological, physical, emotional, social and financial consequences of the direct care of a family member [20]. It this study burden was measured exclusively in caregivers using the Caregiver Burden Inventory (CBI), already validated in this population of HF caregivers [21]. The instrument has five dimensions which aim to detect time-dependent, developmental, physical, social and emotional burden. The items (24) used a 5-point self-report scale, with a score system ranging from 0 (minimum burden) to 4 (maximum burden). A higher score on the CBI means higher burden, with a minimum of 0 and a maximum of 20 per each subscale. Cronbach alpha in our sample was for the whole scale was 0.94. A total score >36 indicates a risk of "burning out" whereas scores near or slightly above 24 indicate a need to seek some form of respite care.

Care dependency was measured with the Care Dependency Scale (CDS) scale [22]. Care dependency is defined as the received support to a patient whose self-care abilities have decreased and whose care demands make him/her to a certain degree dependent [22].

The CDS consists of 15 items—including biopsychosocial needs that every person, whether healthy or ill, has and wants to satisfy—such as eating and drinking, continence, body posture, mobility, day/night pattern, getting dressed and undressed, body temperature, avoidance of danger, hygiene, communication, contact with others, sense of rules and values, daily activities, recreational activities, and learning activities. Responses range from '1 = completely dependent' to '5 = almost independent. In our sample reliability, calculated using Cronbach Alpha,

was 0.98 in patients' sample and 1 in caregivers' sample. In caregivers the proxy version was used to know their point of view about patients 'dependency.

The Montreal Cognitive Assessment tool (MoCA) was chosen for the cognitive assessment in patients and used as a screening tool. Mild cognitive impairment (MCI) is an intermediate clinical state between normal cognitive aging and dementia, and it precedes and leads to dementia in many cases [23]. It is a paper-and-pencil tool that requires approximately 10 minutes to administer and is scored out of 30 points; a score of 26 and higher is considered normal. It assesses multiple cognitive domains, including attention, concentration, executive functions, memory, language, visuospatial skills, abstraction, calculation, and orientation [23]. In our sample reliability, calculated using Cronbach Alpha, was 0.95 in patients' sample.

The SF-8 Health Survey is an 8-item short form survey designed to provide a health-related quality of life (HRQL) profile [24]. The HRQL refers to an individual or group's perception of their overall health, and how it affects their daily life. The instrument uses single-item scales addressing eight domains of general health, physical functioning, role limitations due to both physical health and emotional problems, bodily pain, vitality (energy/fatigue), social functioning, and mental health. The SF-8 generates a health profile of eight discrete scores describing HRQL, which are summarized into physical component (PCS) and mental component (MCS) continuous summary scores. Higher summary PCS and MCS scores indicates better health. The questionnaire was administrated to both caregivers and patients. In our sample reliability, calculated using Cronbach Alpha, was 0.91.

Social support is a multi-faceted concept that positively influences disease-related outcomes in multiple chronic illnesses, including HF. It was defined as the assistance and protection given to others, social support is thought to act as a buffer in stressful situations [25]. Social support has also been described as the exchange of resources between two or more individuals. Social Support (SS) was assessed only in caregivers, to so do we used the Multi-dimensional Scale of Perceived Social Support (MSPSS) which explore the respondent's perception of the support that he/she receives from three diverse sources: a significant other, family, and friends [26]. The MSPSS has twelve items that measure the perceived adequacy of the available amount of SS. The amount of SS is rated on a 7-point Likert scale with responses ranging from very strongly disagree (1) to agree (7) very strongly. The total is calculated through the sum across all the items then divided by 12, with higher scores indicating higher perceived social support. In our sample reliability, calculated using Cronbach Alpha, was 0.93 in caregivers' sample.

## Data analysis

We analysed the quantitative data using descriptive statistics. A multiple linear regression was performed, only for those subjects who reported no missing data for the variables analysed, to predict the CBI using as independent variables those with univariate p-value < 0.05 and supported by literature. The stepwise method was used. Independence of residuals and multi-collinearity were verified via Durbin-Watson and the Variance Inflation Factor (VIF) statistics, respectively. It was estimated that a minimum sample size of 181 patients could achieve 90% power to conduct multiple linear regression analysis to predict CBI using six predictors with a significance level of p < 0.05 and a postponed effect size of 0.10. Sample size was calculated using GPower 3.1. The goodness-of-fit of the regression model was evaluated through the R squared ($R^2$), a global measure of variance. Unstandardized (B) and standardized (β) regression coefficients, standard error (SE), and confidence interval (CI) were considered to describe the models. A p-value of < 0.05 was considered statistically significant.

Qualitative data were analysed using content analysis [27]. An abducting approach was used, stemming from an interactive process of deductive and inductive reasoning. Descriptive

and magnitude coding were chosen as appropriate for the aim of this study. Initially, 98 codes were identified and collected in a codebook, to which another 13 were added during the second-round coding [28]. The mixed methods analysis was completed through simultaneous integration after individual qualitative and quantitative analysis. The merged results were presented in joint displays depicting the inferences and confirmed, discordant, and expanded metainferences [29, 30].

### Ethical considerations

The study complied with the Declaration of Helsinki. Ethics committees at each site approved the research protocol, and patients and caregivers signed informed consents before data collection.

## Results

The quantitative analysis included 184 dyads. Patients' mean age was 73.92 (SD 12.67). The sample was composed of patients who were men (101,54.89%), retired (112, 60.87%), married (112, 60.87%) with NYHA class at least II (89, 74.59%). Caregivers' mean age was 57.57 (SD 14.06). The sample was mainly composed of caregivers who were women (134,72.83%), married (144,78.26%), patients' spouses (79, 43.17%) and in a good economic situation (161, 87.98%) otherwise employed (Table 1).

Descriptive results, reported in Table 2, indicates not worrying levels of burden, even if the caregiver's physical and mental health is worse than those perceived by patients. The patients, however, had mean MoCA values below the cut off of 26, thus registering minimal but present cognitive impairment. Finally, patients seem to perceive themselves as more independent than the caregivers' perspective of them (61.95 vs. 59.32). However, the levels of support perceived by caregivers are quite low (1/3 of the total possible score).

The stepwise multiple linear regression model identified patient MoCA total score ($\beta$ = -0.255; p < 0.001), patient SF-8 physical dimension ($\beta$ = 0.212; p = 0.004), and caregiver CDS total score ($\beta$ = - 0.189; p = 0.009) as independent predictors of CBI. The model explained 24.1% of the variance (Table 3).

### Qualitative findings

A nested sample of 50 caregivers participated in semi-structured interviews to better understand the caregiver experience.

**Sacrificing personal time for care.** Caring required a huge amount of time and dedication from the caregivers. As a result, it took time away from their personal hobbies and activities. Sometimes, the caregivers gave up their time reserved for meeting with peers. Caregivers were forced into social isolation, and young caregivers particularly experienced estrangement from their peer group. The sheer amount of time required to improve the quality of care and life for their relatives with HF affected the entire life pattern and structure of the caregivers' lives. After the onset of the illness, caregivers experienced a total and gradual renovation of their life. Especially at the beginning, caregivers reported that they were deeply involved in caring. Caregivers had to face many difficulties in daily life, even in moments of pause or relaxation. The presence of the illness completed changed their patterns. Furthermore, special occasions were also modified. Heart failure was a constant worry in the dyads' lives, including while on vacation, as reported by this caregiver:

> *"The only limit is on vacation because my husband must always carry the case with the heart monitoring device and therefore, I changed many things compared to before. But I get used to it."* (Italian wife, 61 years old)

**Table 1. Demographic information of the dyads (N = 184).**

| Variable | Patients | | Caregivers | |
|---|---|---|---|---|
| | **Mean** | **SD** | **Mean** | **SD** |
| **Age** | **73.92** | **12.67** | **57.57** | **14.06** |
| | **n** | **%** | **n** | **%** |
| **Gender** | | | | |
| Male | 101 | 54.89 | 50 | 27.17 |
| Female | 83 | 45.11 | 134 | 72.83 |
| **Marital status** | | | | |
| Married | 112 | 60.87 | 144 | 78.26 |
| Widower | 59 | 32.07 | 2 | 1.09 |
| Single | 9 | 4.89 | 32 | 17.39 |
| Divorced | 4 | 2.17 | 6 | 3.26 |
| **Employment** | | | | |
| Retired | 142 | 77.6 | 49 | 26.78 |
| Full-time employee | 15 | 8.2 | 43 | 23.50 |
| Part-time employee | 9 | 4.92 | 9 | 4.92 |
| Housewife | 8 | 4.37 | 29 | 15.85 |
| Unemployed in order to take care of the patient | 6 | 3.28 | 9 | 4.92 |
| Unemployed | 2 | 1.09 | 19 | 10.38 |
| Freelance | 1 | .55 | 25 | 13.66 |
| **Patient NYHA class** | | | | |
| Class I | 30 | 25.21 | - | - |
| Class II | 53 | 44.54 | - | - |
| Class III | 36 | 30.25 | - | - |
| **Economic situation** | | | | |
| I have what I need to live | - | - | 161 | 87.98 |
| I have more than I need to live | - | - | 13 | 7.10 |
| I don't have what I need to live | - | - | 9 | 4.92 |
| **Caregiver lives with the patient (yes)** | - | - | 113 | 61.41 |
| **Relationship with the patient** | | | | |
| Spouse | - | - | 79 | 43.17 |
| Son/daughter | - | - | 50 | 27.32 |
| Son/daughter-in-law | - | - | 31 | 16.94 |
| Brother/sister | - | - | 8 | 4.37 |
| Friend | - | - | 4 | 2.19 |
| Other | - | - | 11 | 6.01 |

**Anticipating bleak future.** Future dreams, expectations, and life goals projected for themselves were set aside and the care of their loved ones was prioritized. Caregivers noted they felt the future seemed unlikely to improve. Women reported being forced to give up their jobs completely, while men were more likely to make drastic adjustments. Caregivers, directly or in an implicit way, reported that their lives changed for the worse after role-taking. They felt pessimistic about the future and shared uncertainties and fears.

*"I can't imagine my life... for others I'm more positive, for me I don't know. I live for the day. I would like to think about a future but there is not one. If it gets worse, I'll have other types of needs, if it if it goes on like this, no."* (Italian woman, age 40)

**Table 2. Descriptive results of burden and related aspects (N = 184).**

| Variable | Patients | | Caregivers | |
|---|---|---|---|---|
| | **Mean** | **SD** | **Mean** | **SD** |
| MoCA total | 23.72 | 5.24 | - | - |
| SF8 physical dimension | 12.09 | 3.96 | 8.36 | 3.69 |
| SF8 mental dimension | 10.21 | 3.53 | 8 | 3.35 |
| CDS total | 61.95 | 16.02 | 59.32 | 16.49 |
| MSPSS total | - | - | 9.66 | 7.34 |
| CBI total | - | - | 14.86 | 15.48 |

Note: Caregiver Burden Inventory (CBI); Care Dependency Scale (CDS); Montreal Cognitive Assessment tool (MoCA); SF8: Health-related quality of life (HRQL) profile; Multi-dimensional Scale of Perceived Social Support (MSPSS)

**Somatic stress.** The caregivers explained that they had poor physical health entailing pain, generalized fatigue, and exhaustion. They experienced many somatic symptoms due to mental stress caused by being hyperalert and apprehensive of their loved ones. Even most nights, they did not experience a moment of pause or rest from their duties and therefore experience daily fatigue. Referring to the strain associated with the demands of caregiving, they experienced health disorders. Particularly they reported having low physical health, as illustrated by one of the caregivers:

> *"Bad mood, stress, anxiety, stomachache, not sleeping, waking up every time the phone rings, running around the house, yes, that's all... Everything... Everything. Pain in my knees, it hits me... It hits my knees."* (Spanish man, age 52)

**Isolation.** Caregivers who needed help were removed by their families. Women in particular, who had to deal with several family members depending on them (not necessarily suffering from pathologies, e.g., children) paid the price of their multiple roles by experiencing greater stress. Family problems were also found in the marital relationship. At a social and economic level, this resulted in the abandonment of work or a reduction in productivity hours. When family support was lacking, caregivers experienced a sense of marginalization and abound from the other family members often resulting in definitive breaks. A caregiver reported:

> *"I thought that being the youngest, my sisters would be closer to me; I have not seen anyone. They moved away despite knowing about the disease, but they ignored it. They only approached during the funeral of a relative but then no continuation. It was not wanted by*

**Table 3. Stepwise multiple linear regression of the study variable on caregiver burden ($R^2$ = 0.241).**

| Predictor | B (SE) | β | *p*-value | 95% CI |
|---|---|---|---|---|
| Patient MoCA total score | - .754 (.209) | - .255 | < .001 | - 1.165 to—.342 |
| Patient SF-8 physical dimension | .829 (.281) | .212 | .004 | .275 to 1.383 |
| Caregiver CDS total score | - .178 (.067) | - .189 | .009 | - .311 to—.045 |

Variables excluded from the final model: patient age, patient SF-8 mental dimension, caregiver MSPSS total score.

**B** = unstandardized regression coefficient; **SE** = standard error; **β** = standardized regression coefficient; **CI** = confidence interval

*me. I saw myself abandoned in such a delicate moment and I closed the chapter. Maybe they thought I was a burden to them and so I said stop.*" (Italian woman, age 58)

**Worry in time.**  Caregivers who experienced temporally oriented feelings, such as guilt and helplessness, felt they were not doing enough for their loved one. Helplessness was also perceived to be past-oriented if the caregiver compared their loved one's current state with the past before the onset of the disease and old age. The current sense of pity and bitterness toward their loved ones due to the pathological conditions caused the caregivers to have feelings of encouragement to do better for the future. Patients' care required continuous attention and the caregivers were constantly focused on the patients, so they developed a continuous and constant sense of apprehension towards their loved ones:

*"He has changed our family habits a little, so there is more apprehension towards him. We used to lead a little bit more lively life now we lead a little bit quieter life. We are more careful about what we do because we don't want the same thing to happen again. And then last year he had an episode of tachycardia of rapid heartbeats, he was hospitalized, and we had a lot of apprehension again and we worried."* (Italian woman, age 48)

**Prioritization of patients' health.**  The high level of care dependence of the patient modified how caregivers lived their lives. Some patients were totally dependent on the caregiver in daily life activities, and this modified the pattern of the structure of the organization of the day because the caregivers had to prioritize the patient's needs. Although not all patients were highly dependent, the risk of re-exacerbation of the illness led the dyads to always remain by their side. One of the caregivers noted:

"*He fainted, everywhere where he is walking, help is needed, he cannot go to hospital independently.*" (Dutch wife, 55 years old)

**Repetitive demands.**  The caregivers found the outcomes of MCI on the patient stressful. Cognitive impairment led to repetitive demands for care throughout the day which then forced the caregiver to be at the patient's constant disposal. In cases where the patient lacked memory and attention, caregiver-patient conversations were reported as if the patient was telling the same story repeatedly, thereby affecting the quality of communication. One of the caregivers reported:

"*So I tell him, "Come on, \*\*\*\*, you can do it yourself!" But no, he is not able to do it. . .So I have to take him to the bathroom. And when he's ready, he calls me again, "Can you pull up my pants?" I say: "Come on \*\*\*\*, you can do it yourself!". It's not possible, come on! it's exhausting! And often 6 times in one evening I always have to take him to the bathroom. The same thing happened to get dressed, we tried it last night. He always wears a T-shirt and his clothes from above. But hey! he is not able to tuck his shirt into his pants so I have to come and take his pants off, take his shirt off push it back in, put his pants back on and so even 6 times in an evening. And then I didn't say that this happens all day every day.*" (Dutch wife, 55 years old)

**Restructuring life.**  Caregivers reported—especially if they were living with the patient—that dedicating oneself to patient care was a full-time commitment. Patients with a high level of dependence were unable to perform activities of daily living. Caregivers deeply felt the weight of their irreplaceable role in the patient's life. Caregivers restructured their lives in

| Quantitative Data | Qualitative Data | Mixed Methods Metainferences |
|---|---|---|
| **Inference:** Caregiver experienced burden across all the dimensions, but greater burden was perceived in terms of time, development, and physical dimensions. | *"Sacrificing Personal Time for Care,"* *"Worry in time,"* *"Anticipating Bleak Future,"* and *"Somatic Stress"* were consistent with the highest scored burden dimensions. Caregivers noted that time issues, limited outlook, fatigue, and exhaustion greatly affected their abilities to care for their loved ones. | Time constraints, anticipated concerns about personal future, and generalized fatigue and exhaustion were significant predictors of caregiver burden, thereby affecting their caregiving. |

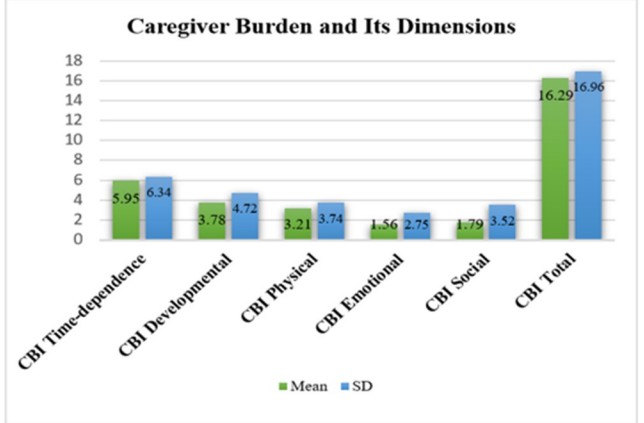

**Fig 1. Joint display of caregiver burden and its dimensions.**

terms of sleeping habits, eating patterns, social functioning, and communication. In some cases, the commitment required was so high that patients needed it even during the night. An Italian caregiver stated:

> "*My father was always there and could handle so many things. Anything that I couldn't manage. Dad was always there to think about helping me. And now that is no longer the case. If my father used to help me and my children do things now I have to do everything by myself. I had to relearn how to do everything without help.*" (Italian daughter, 55 years old)

**Mixed methods analysis.** During integration the qualitative themes were compared with the various domains of quantitative scales resulting in confirmed and expanded findings. Confirmed findings were those in which qualitative and quantitative data were consistent. However, expanded findings were those which provided additional understanding into the quantitative results. The first display presents a comparative analysis of the means scores of burden, quality of life, perceived SS, and cognitive impairment (Fig 1). The second display offers a comparison of regression analysis and the qualitative themes and illustrates the confirmed findings (Table 4). However, several additional predictors of caregiver burden were identified through qualitative exploration, and these predictors were labelled as expanded findings. The additional predictors were *sacrificing personal time for care*, *anticipating bleak future*, *somatic stress*, *worry in time*, and *restricting life*. These themes were consistent with the domains of caregiver burden (Fig 1).

## Discussion

This study provides a comprehensive understanding of caregiver burden and its predictors from a dyadic perspective. Qualitative and quantitative analysis demonstrated that caregiver burden was predicted by patients' worse cognitive impairment, lower physical quality of life, and the degree of care dependency perceived by caregivers. To our knowledge, this is the first study that noted caregiver dependency as a predictor of caregiver burden from the perspective

**Table 4. Joint display of predictors of caregiver burden.**

| Quantitative Inferences | Qualitative Inferences | Mixed Methods Metainferences |
|---|---|---|
| Patients' cognitive impairment was a negative predictor [MoCA total score (β = - 0.255; p < 0.001)] | Caregivers note that *"Repetitive Demands"* of care resulted in stress and affected their ability to provide effective care. One of the caregivers noted: *"I have to bring a chair every evening where he can scramble in and out. It is really hard because (yes) then I am reading or watching television and (yes) then he has to pee, and I have to get up again. And (yes), that's heavy. But (yes), I can't have someone around us all day long who does that."* (Dutch woman, age 78) | **Confirmed.** Cognitive impairment was a significant predictor of caregiver burden. Increased impairment led to communication issues that aggravated the overall caring interaction with the caregiver and the loved one with HF. |
| The physical dimension of quality of life was a predictor [SF-8 physical dimension (β = 0.212; p = 0.004)] | The theme "Prioritization of Patients' Health" tapped into the physical dimension of qualify of life. Caregivers discussed that they prioritized the needs and care of their loved ones over their necessities. One of the caregivers noted: *"You know? My health, well, I think. . . that is not. . . It's not bad, but I'm fifty-two years old, my back is not particularly good anymore, and well, I take better care of others than I take care of myself."* (Spanish woman, age 52) | **Confirmed.** The physical dimension of the quality of life scale was a significant predictor of caregiver burden. Caregivers neglected their needs and focused on providing better care for their loved ones. |
| Patient care dependency was a predictor [caregiver CDS total score (β = - 0.189; p = 0.009)] | Increased and repetitive patient demands and needs pertaining to eating and drinking, mobility, dressing, and dependence on caregivers affected the caregiving process. One of the caregivers noted: *"I always have to keep an eye on him, he forgets about the medicines, I tell him- you don't have to drink wine, the doctor has forbidden it-, we discuss a little, then I tell him not to drink anymore. . . ."* (Italian woman, age 77) | **Confirmed.** The care dependency of patients with HF affected the ability of caregivers to engage in the provision of quality care. |

of informal caregivers. In line with previous studies [8], this study also noted that informal caregivers experienced isolation and prioritize the patients' needs even at the expense of their own health. Therefore, it is imperative that programs and interventions are developed to support caregivers in their caregiving process. Health care professionals should collaborate with the caregivers and prepare them to address the challenges associated with caregiving. Further research is warranted to design and evaluate caregiver support programs to address their burden in community and home care settings across different cultural contexts.

The results find that the duration of caregiving and perceived physical and emotional stress results in longevity of caregiver burden. Consistently, a recent concept analysis of 33 nursing literature sources also identified that caregiver burden is time-bound and contingent on the caregivers' physical and emotional state and quality of life [31]. The longevity of caregiving affecting caregiver physical and emotional stress can have detrimental effect on the well-being and care of their loved ones with illness. Therefore, the caregiver support programs may include provision of formal caregiving resources, emotional health assessments, and frequent consultations with health care professionals to combat their burden. These findings have implications for social policy making concerning adequate and timely launch and provision of governmental and community based programs to support informal caregivers in their work. For example, policy development could focus on providing financial, emotional, and formal caregiving resources to informal caregivers.

This study provides an overview of caregiver burden and its predictors in short term. Further longitudinal studies are required to examine the changes in caregiver burden and the role of these predictors over time. Nevertheless, the knowledge gained about the predictors of caregiver burden is critical for health care providers—particularly nurses and hospital and

community organizations—so that caregivers are adequately prepared to tackle the complexities of caregiving and combat personal stress. Hospitals and nurses could prepare informal caregivers through effective discharge teaching and health education about approaches to manage factors impeding their caregiving abilities. Additionally, the findings substantiate the need for developing integrated and collaborative care involving health care professionals, informal caregivers, formal caregivers, and community based patient and home care organizations to launch programs to educate and train informal caregivers in the community settings.

## Limitations

There are certain limitations in this study. First, we used a convenience sample. Second, it is a secondary analysis from a previous study that clearly has a somewhat different scope and purpose. Nevertheless, the robustness of the secondary data analysis is enhanced by providing a transparent overview of methods of data collection and analysis and a combination of qualitative and quantitative perspectives. Third, for the qualitative phase, the interview guide was not built for the specific purpose declared in this study. Finally, even this study offers the European view of the perspective of caregivers of patients with HF, and therefore results will likely be difficult if applied to contexts with different cultural backgrounds.

## Conclusions

Caregiver burden is a multi-dimensional construct influenced by an array of patient- and caregiver-related factors. Among the dyads, the patient-related predictors included cognitive impairment and level of care dependency. However, the caregiver-related predictors were sacrificing personal time for care, anticipated bleak future, somatic stress, worry in time, and restructuring life. Increased caregiver burden can affect the capability of informal caregivers to support and care for their relatives with HF. There is a dire need for development and evaluation of individual and community-based strategies to address caregiver burden to enhance the quality of life of both caregivers and their relatives with HF.

## Supporting information

**S1 Checklist. Good Reporting of A Mixed Methods Study (GRAMMS) checklist.**
(DOCX)

**S2 Checklist. STROBE statement—Checklist of items that should be included in reports of *cross-sectional studies*.**
(DOC)

## Acknowledgments

The authors affirm that the methods used in the data analyses are suitably applied to their data within their study design and context, and the statistical findings have been implemented and interpreted correctly.

The authors agree to take responsibility for ensuring that the choice of statistical approach is appropriate and is conducted and interpreted correctly as a condition to submit to the Journal Frequently used statistical methods (descriptive, graphical methods, parametric & non-parametric tests, linear & logistic regression).

**Patient or Public Contribution**: Caregivers were involved after they had been observed in their dynamics of involvement in caring of the familiars or friends with heart failure. They were recruited after the discharge or at the time of the outpatient visit.

Institutions where the work was performed: University of Rome "Tor Vergata", Rome, Italy; Hospital Lozano Blesa, Zaragozza, Spain and Maastricht University Medical Center, MUMC, Patiënt & Zorg, Maastricht, The Netherlands.

## Author Contributions

**Conceptualization:** Angela Durante, Ahtisham Younas, Raul Juarez-Vela, Ercole Vellone.

**Data curation:** Angela Durante.

**Formal analysis:** Angela Durante, Ahtisham Younas, Angela Cuoco, Josiane Boyne, Bridgette M. Rice, Raul Juarez-Vela, Valentina Zeffiro, Ercole Vellone.

**Funding acquisition:** Angela Durante, Ercole Vellone.

**Investigation:** Angela Durante, Josiane Boyne, Bridgette M. Rice, Valentina Zeffiro, Ercole Vellone.

**Methodology:** Angela Durante, Ahtisham Younas, Angela Cuoco, Josiane Boyne, Bridgette M. Rice, Raul Juarez-Vela, Ercole Vellone.

**Resources:** Angela Durante, Raul Juarez-Vela, Valentina Zeffiro.

**Supervision:** Josiane Boyne, Raul Juarez-Vela, Ercole Vellone.

**Validation:** Ahtisham Younas, Ercole Vellone.

**Writing – original draft:** Angela Durante, Ahtisham Younas, Angela Cuoco, Josiane Boyne, Bridgette M. Rice, Raul Juarez-Vela, Valentina Zeffiro, Ercole Vellone.

**Writing – review & editing:** Angela Durante, Ahtisham Younas, Josiane Boyne, Bridgette M. Rice, Raul Juarez-Vela, Valentina Zeffiro, Ercole Vellone.

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
