## [Decision Letter · Decision Letter 0]

17 Jul 2023

PONE-D-23-14407Burden Among Informal Caregivers of Individuals with Heart Failure: A Mixed Methods StudyPLOS ONE

Dear Dr. Younas,

Thank you for submitting your manuscript to PLOS ONE. After careful consideration, we feel that it has merit but does not fully meet PLOS ONE’s publication criteria as it currently stands. Therefore, we invite you to submit a revised version of the manuscript that addresses the points raised during the review process.

We look forward to receiving your revised manuscript.

Kind regards,

Giuseppe Marano

Academic Editor

PLOS ONE

“The first author was funded by the Heart Failure Association with the Nursing Training Fellowship Award. The project received funding also from the CECRI Centre of Excellence for Nursing Scholarship in Rome (2.12.5).”

Reviewers' comments:

Reviewer's Responses to Questions

**Comments to the Author**

1. Is the manuscript technically sound, and do the data support the conclusions?

Reviewer #1: Yes

Reviewer #2: Yes

2. Has the statistical analysis been performed appropriately and rigorously? 

Reviewer #1: Yes

Reviewer #2: Yes

3. Have the authors made all data underlying the findings in their manuscript fully available?

Reviewer #1: Yes

Reviewer #2: Yes

4. Is the manuscript presented in an intelligible fashion and written in standard English?

Reviewer #1: Yes

Reviewer #2: Yes

5. Review Comments to the Author

Reviewer #1: Thank you for the opportunity to review “Burden Among Informal Caregivers of Individuals with Heart Failure: A Mixed Methods Study”. The topic is interesting as well as the manuscript itself. Many correlates of caregiver burden are already known, but there is still a lack of qualitative studies exploring the nature of caregiver burden. This article gives an insight into the emotional processes of the dyad and this is its great value.

Although the article is well organized there are a few comments that may improve it:

I suggest including mutuality, which can be defined as the degree of emotional engagement and mutual support in the dyad, as an additional crucial feature in the Introduction section.

(see i.e. Uchmanowicz I, Faulkner KM, Vellone E, Siennicka A, Szczepanowski R, Olchowska-Kotala A. Heart Failure Care: Testing Dyadic Dynamics Using the Actor-Partner Interdependence Model (APIM)-A Scoping Review. Int J Environ Res Public Health. 2022 Feb 9;19(4):1919. doi: 10.3390/ijerph19041919. PMID: 35206131; PMCID: PMC8871794. ; Hooker SA, Schmiege SJ, Trivedi RB, Amoyal NR, Bekelman DB. Mutuality and heart failure self-care in patients and their informal caregivers. Eur J Cardiovasc Nurs. 2018 Feb;17(2):102-113. doi: 10.1177/1474515117730184. Epub 2017 Sep 4. PMID: 28868917; PMCID: PMC9390005.)

It would also be advisable to include in the introduction the work of Strömberg (Strömberg, A.; Luttika, M.L. Burden of Caring: Risks and Consequences Imposed on Caregivers of Those Living and Dyingwith Advanced Heart Failure. Current Opinion in Supportive and Palliative Care. 2015, pp 26–30.

https://doi.org/10.1097/SPC.0000000000000111, who draws attention to having an uncertain HF trajectory that places a significant burden on heart failure caregivers.

Reviewer #2: Comments to the Authors

Thank you for providing the opportunity to review the manuscript “Burden Among Informal Caregivers of Individuals with Heart Failure: A Mixed Methods Study”. The aim of this study was to develop a comprehensive understanding of caregiver burden and its predictors from a dyadic perspective. The followings are the comments.

Introduction:

1. It would be better to make some expansion on the concerning variables that have been chosen for exploration alongside caregiver burden.

2. The author repeatedly emphasized the lack of "Cross-cultural studies" in this area, and the sample was from three different countries. However, cross-cultural differences and analysis were not reflected in the current study.

3. Considering the accumulation of a large amount of literature related to the caregiver burden, the description of this sentence “Furthermore, there is a dearth of qualitative studies exploring the nature of caregiver burden and any additional predictors that may affect the caregiver burden” need to be cautious.

Methods

1. In Data Collection Instruments, please specify in which sample SF-8 was used, patients, caregivers, or both?

2. Considering that interview guide of the qualitative phase was not built for the specific purpose declared in this study, it would be better to list a few interview questions for understanding easier.

3. In Setting and Sample, the sample size is 229, it does not match the 184 reported in the results section. How many caregivers are there to participate in the qualitative research?

Results

1. In the first paragraph, the description of frequency and percentage is incorrect (cannot use “±” and SD). Additionally, the frequency and percentage of patients' marital status does not match that in the Table1. Please correct it.

2. Qualitative Findings-Repetitive Demands, the summary of this topic does not align with the example listed, please provide a new one to match better.

3. Page 15, line 1-2: the “restricting life” is inconsistent with the topic title “Restructuring Life”.

6. PLOS authors have the option to publish the peer review history of their article (what does this mean?). If published, this will include your full peer review and any attached files.

Reviewer #1: No

Reviewer #2: No

---

## [Author Response · Author response to Decision Letter 0]

15 Sep 2023

A point-by-point response has been submitted

---

## [Decision Letter · Decision Letter 1]

3 Oct 2023

Burden Among Informal Caregivers of Individuals with Heart Failure: A Mixed Methods Study

PONE-D-23-14407R1

Dear Dr. Younas,

We’re pleased to inform you that your manuscript has been judged scientifically suitable for publication and will be formally accepted for publication once it meets all outstanding technical requirements.

Kind regards,

Giuseppe Marano

Academic Editor

PLOS ONE

Additional Editor Comments (optional):

Reviewers' comments:

Reviewer's Responses to Questions

**Comments to the Author**

1. If the authors have adequately addressed your comments raised in a previous round of review and you feel that this manuscript is now acceptable for publication, you may indicate that here to bypass the “Comments to the Author” section, enter your conflict of interest statement in the “Confidential to Editor” section, and submit your "Accept" recommendation.

Reviewer #1: All comments have been addressed

Reviewer #2: All comments have been addressed

2. Is the manuscript technically sound, and do the data support the conclusions?

Reviewer #1: Yes

Reviewer #2: Yes

3. Has the statistical analysis been performed appropriately and rigorously? 

Reviewer #1: Yes

Reviewer #2: (No Response)

4. Have the authors made all data underlying the findings in their manuscript fully available?

Reviewer #1: Yes

Reviewer #2: (No Response)

5. Is the manuscript presented in an intelligible fashion and written in standard English?

Reviewer #1: Yes

Reviewer #2: (No Response)

6. Review Comments to the Author

Reviewer #1: (No Response)

Reviewer #2: Thank you for providing the opportunity to review the manuscript “Burden Among Informal Caregivers of Individuals with Heart Failure: A Mixed Methods Study”. In the revised version, the authors have been responsive to the issues raised.

7. PLOS authors have the option to publish the peer review history of their article (what does this mean?). If published, this will include your full peer review and any attached files.

Reviewer #1: No

Reviewer #2: No

---

## [Editor Report · Acceptance letter]

6 Oct 2023

PONE-D-23-14407R1 

Burden Among Informal Caregivers of Individuals with Heart Failure: A Mixed Methods Study 

Dear Dr. Younas:

I'm pleased to inform you that your manuscript has been deemed suitable for publication in PLOS ONE. Congratulations! Your manuscript is now with our production department. 

Kind regards, 

on behalf of

Dr. Giuseppe Marano 

Academic Editor

PLOS ONE